# Nivolumab after Induction Chemotherapy in Previously Treated Non-Small-Cell Lung Cancer Patients with Low PD-L1 Expression

**DOI:** 10.3390/cancers15184460

**Published:** 2023-09-07

**Authors:** Beung-Chul Ahn, Charny Park, Sang-Jin Lee, Sehwa Hong, Ji-Eun Hwang, Kyoungsuk Kwon, Jin Young Kim, Kyung-Hee Kim, Hyae Young Kim, Geon Kook Lee, Youngjoo Lee, Ji-Youn Han

**Affiliations:** 1Center for Lung Cancer, Division of Hematology and Oncology, Department of Internal Medicine, Research Institute and Hospital, National Cancer Center, Goyang-si 10408, Gyeonggi-do, Republic of Korea; abcduke@yuhs.ac (B.-C.A.); kks17@ncc.re.kr (K.K.);; 2Research Institute, National Cancer Center, Goyang-si 10408, Gyeonggi-do, Republic of Korea; charn78@ncc.re.kr (C.P.); leesj@ncc.re.kr (S.-J.L.); jyhwang@gmail.com (J.-E.H.); jykim@ncc.re.kr (J.Y.K.); 3Proteomics Core Facility, Research Core Center, Research Institute and Hospital, National Cancer Center, Goyang-si 10408, Gyeonggi-do, Republic of Korea; 4Department of Radiology, Research Institute and Hospital, National Cancer Center, Goyang-si 10408, Gyeonggi-do, Republic of Korea; radhykim@ncc.re.kr; 5Department of Pathology, Research Institute and Hospital, National Cancer Center, Goyang-si 10408, Gyeonggi-do, Republic of Korea; gklee@ncc.re.kr

**Keywords:** nivolumab, cyclophosphamide, adriamycin, induction chemotherapy, transferrin receptor protein

## Abstract

**Simple Summary:**

Different strategies have been explored to counteract immune evasion by shifting the balance in favor of antitumor immune activation, and combination cytotoxic chemotherapies have emerged as potent immune modulators for patients with low programmed death-ligand 1 expression. This study aimed to investigate whether the addition of cyclophosphamide and adriamycin induction therapy prior to nivolumab could enhance the efficacy of immune checkpoint inhibitors in patients previously treated with non-squamous non-small-cell lung cancer with less than 10% programmed death-ligand 1 expression. Patients with a durable response to nivolumab showed higher baseline transferrin receptor protein levels. The predictive role of transferrin receptor protein as a biomarker for immune checkpoint inhibitors in non-squamous non-small-cell lung cancer with low programmed death-ligand 1 expression was validated in an independent cohort.

**Abstract:**

This study aimed to investigate whether cyclophosphamide (C) and adriamycin (A) induction therapy (IT) prior to nivolumab could enhance the efficacy of nivolumab in previously treated patients with non-squamous (NSQ) non-small-cell lung cancer (NSCLC) with less than 10% programmed death-ligand 1 (PD-L1) expression. Twenty-two enrolled patients received four cycles of CA-IT every 3 weeks. Nivolumab was given 360 mg every 3 weeks from the second cycle and 480 mg every 4 weeks after four cycles of CA-IT. The median progression-free survival (PFS) and overall survival (OS) were 2.4 months and 11.6 months, respectively. Fluorescence-activated cell sorting revealed the lowest ratio of myeloid-derived suppressor cells (MDSCs) to CD8+T-cells in the responders. Proteomic analysis identified a consistent upregulation of extracellular matrix-receptor interactions and phagosome pathways in the responders. Among the differentially expressed proteins, the transferrin receptor protein (TFRC) was higher in the responders before treatment (fold change > 1.2). TFRC validation with an independent cohort showed the prognostic significance of either OS or PFS in patients with low PD-L1 expression. In summary, CA-IT did not improve nivolumab efficacy in NSQ-NSCLCs with low PD-L1 expression; however, it induced decreasing MDSC, resulting in a durable response. Higher baseline TFRC levels predicted a favorable response to nivolumab in NSCLC with low PD-L1 expression.

## 1. Introduction

Lung cancer is a significant cause of cancer-related deaths worldwide, and non-small-cell lung cancer (NSCLC) accounts for approximately 85% of all lung cancer cases [1]. Nowadays, programmed cell death protein 1 (PD-1)/programmed death-ligand 1 (PD-L1) immune checkpoint inhibitors (ICI), which unleash T lymphocyte-mediated immune responses, have become a breakthrough therapy for lung cancer. The CheckMate 057 trial is a clinical trial that evaluated the efficacy and safety of nivolumab, a PD-1 programmed death-ligand inhibitor, compared to docetaxel in patients with previously treated advanced NSCLC. The trial showed that nivolumab improved overall survival and had a better safety profile than docetaxel, leading to its approval by the FDA as a treatment option for patients with advanced NSCLC who had previously received platinum-based chemotherapy.

Currently, immunohistochemical-based ((IHC)-based) PD-L1 expression is the only predictive marker approved for ICIs, reflecting the immune characteristics of the tumor to escape immune surveillance. According to research on the early survival of patients in the CheckMate 057 trial, patients with poor prognostic factors combined with no or low PD-L1 expression were at a higher risk of death within the first 3 months of nivolumab treatment than with docetaxel [2]. In the same context, in Korea, reimbursement for nivolumab is limited to patients with PD-L1 expression levels ≥ 10% and who have progressed to platinum-based chemotherapy due to its better hazard ratio among subgroups with PD-L1 expression levels ≥ 10%. However, PD-L1 expression alone may not comprehensively reflect the complexity of the tumor microenvironment (TME), and other immunosuppressive mechanisms within the TME may attenuate ICI response. Collectively, these mechanisms contribute to the low mutational burden, poor intrinsic antigenicity of tumor cells, absence of priming, defective antigen presentation during the priming phase, and functional exhaustion of tumor-infiltrating lymphocytes by suppressive immune regulatory cells [3].

Different strategies have been explored to counteract immune evasion by shifting the balance in favor of antitumor immune activation, and combination cytotoxic chemotherapies have been used as potent immune modulators [4]. So far, several published studies have demonstrated the beneficial effect of low-dose cyclophosphamide in amplifying the immune response against tumors by reducing regulatory T cells (Tregs) [5]. Doxorubicin is considered a good immunomodulator as it induces CD8^+^ IFN-γ^+^ T cell responses and significantly upregulates T cell activation markers on the surface of CD4^+^ T cells. In addition, it promotes cytokine secretion of interleukin (IL)-1, IL-2, and tumor necrosis factor α (TNF-α), enhancing the immune shift from Th2 to Th1 [6,7]. Furthermore, doxorubicin selectively impairs myeloid-derived-suppressor-cell (MDSC)-induced immunosuppression [8]. These two drugs are conventional cytotoxic drugs with good accessibility and cost benefits. Herein, we comprehensively assessed the impact of cyclophosphamide and Adriamycin (CA) induction therapy on the antitumor effects of nivolumab in advanced NSCLC with PD-L1 < 10%.

## 2. Materials and Methods

### 2.1. Study Design

This was a single-center, single-arm, phase II study. The patients received 4 cycles of cyclophosphamide (500 mg/m^2^) and adriamycin (50 mg/m^2^) induction therapy every 3 weeks. Nivolumab 360 mg was started from the second cycle of the induction phase. After induction therapy, nivolumab 480 mg was administered every four weeks until disease progression, unacceptable toxicity, or for a maximum of two years. The study was approved by the research ethics board (IRB No. NCC2018-0267) and conducted in accordance with the ethical principles of the Declaration of Helsinki. This study was registered at clinicaltrials.gov (NCT03808480), and written informed consent was obtained from all the patients.

### 2.2. Participants

To be eligible for the study, all patients aged 18 years or over were required to have histological or cytological proof of advanced non-squamous NSCLC. Patients with measurable lesions were eligible for inclusion. Each patient was required to have wild-type epidermal growth factor receptor (EGFR), anaplastic lymphoma kinase (ALK), a PD-L1 expression less than 10%, and a performance status of 1 or lower on the Eastern Cooperation Oncology Group (ECOG) scale with adequate organ function. All the patients previously received platinum-based chemotherapy. These patients should have received fewer than three chemotherapy regimens. The main exclusion criteria were a history of malignancies other than lung cancer in the past three years, previous ICI therapy, a history or current diagnosis of interstitial lung disease, and any unstable systemic disease.

### 2.3. Assessment of Response

The patients underwent chest and abdominal computed tomography (CT), brain magnetic resonance imaging (preferred), or CT and positron emission tomography at baseline. Assessments were performed every 2 cycles of therapy and every 2 months thereafter during the long-term follow-up. Using these imaging techniques, the clinical response to nivolumab was assessed in accordance with the Response evaluation criteria in solid tumors version 1.1. The following terms were used to describe the manner in which a tumor responded to treatment: complete response (CR; disappearance of all target lesions), partial response (PR; ≥30.0% reduction in the sum of the diameters of the target lesions), progressive disease (PD; ≥20.0% increase in the sum of the diameters of the target lesions), and stable disease (SD; insufficient to qualify as PR or PD). The objective response rate (ORR) was the proportion of patients who achieved CR or PR, and the disease control rate (DCR) was the proportion of patients who achieved CR, PR, or SD. Progression-free survival (PFS) was defined as the time from assignment to the study until the date of disease progression or death from any cause, whichever occurred first. Overall survival (OS) was defined as the time from the start of the study to the date of death or last follow-up for patients who were alive at the end of the study. The data of patients alive at the last follow-up visit were censored. Adverse events (AEs) were summarized according to the National Cancer Institute Common Terminology Criteria for Adverse Events (version 5.0).

### 2.4. Immunohistochemistry of PD-L1 Expression

PD-L1 expression was determined using a Ventana PD-L1 SP263 antibody (Ventana Medical Systems, Tucson, AZ, USA). PD-L1 expression levels in tumor cells were determined by the percentage of stained cells in each slide, which was estimated in increments of 5%, except for a 1% positivity value. Patients in whom at least 1% of the tumor cells stained positive for PD-L1 were determined positive.

### 2.5. Blood Next-Generation Sequencing Analysis (NGS) by GuardantOMNI

DNA extraction and next-generation sequencing were subsequently performed at Guardant Health using the Guardant OMNI (Redwood City, CA, USA) 2.15 Mb, 500 gene panel) to identify SNVs, indels, gene fusions, copy-number variants, microsatellite status, and tumor mutation burden (TMB) [9]. The GuardantOMNI algorithm was used to report plasma TMB. This algorithm includes all somatic synonymous and nonsynonymous single-nucleotide variants (SNVs) and indels, excluding germline, clonal hematopoiesis of indeterminate potential, driver, and resistance variants, with statistical adjustment for sample-specific tumor shedding of ctDNA and coverage [10]. pTMB-unevaluable samples are those with limited tumor shedding, including all somatic mutations <0.3% of the maximum somatic allele fraction, or low unique molecule coverage.

### 2.6. Proteomics

#### 2.6.1. Sample Preparation for Mass Spectrometry

Plasma samples were depleted using High Select^TM^ top14 abundant protein depletion spin columns (Thermo Fisher Scientific, Rockford, IL, USA) and digested using S-Trap™ spin columns (Protifi, Huntington, NY, USA) according to the manufacturer’s instructions. Desalted peptides were labeled using the TMT reagent (Thermo Fisher Scientific), and 24 noncontinuous peptide fractions were separated using an Agilent 1260 Infinity HPLC system (Agilent, Palo Alto, CA, USA). Five percent of each fraction was aliquoted for global proteome analysis, and the remaining 95% was combined into 12 fractions for phosphoproteome analysis. Enriched phosphopeptides were obtained by combining 12 fractions with Ni-NTA agarose beads (Qiagen, Valencia, CA, USA) and converting them to Fe^3+^-NTA beads.

#### 2.6.2. LC-MS/MS Analysis

The peptides prepared for global/phosphoproteome analysis were resuspended in 0.1% formic acid in water and analyzed using a FAIMS Pro interface (Thermo Fisher Scientific) mounted on an Orbitrap Eclipse Tribrid mass spectrometer (Thermo Scientific) equipped with an Ultimate 3000 RSLCnano system (Thermo Scientific). Solvents A and B contained 0.1% formic acid (FA) in water and 0.1% FA in acetonitrile, respectively. Peptides were loaded onto a trap column (Acclaum PepMap^TM^ 100, 75 mm × 2 cm) and separated on an analytical column (EASY-Spray column, 75 mm × 50 cm; Thermo Fisher Scientific). Three CVs (−40/−60/−80) were used within a total MS cycle time of 6 s. Full MS scans were acquired over the range *m*/*z* of 350–2000 with a mass resolution of 120,000 (at *m*/*z* 200). Tandem mass spectra were acquired using an Orbitrap mass analyzer with a mass resolution of 30,000 at *m*/*z* 200 using the TurboTMT feature.

#### 2.6.3. Protein Identification and Quantitation

A database search of all raw data files was performed using the Proteome Discoverer 2.5 software (Thermo Fisher Scientific). SEQUEST-HT was used to search the SwissProt Human database. The database search parameters included a precursor ion mass tolerance of 10 ppm, fragment ion mass tolerance of 0.02 Da, static modifications for carbamidomethylation (+57.021 Da/C) and TMT tags (+229.163 Da/K and N-terminal), and variable modifications for oxidation (+15.995 Da/M) and phosphorylation (+79.966 Da/S, T, Y). We obtained an FDR of less than 1% at the peptide level and filtered with high peptide confidence. The reporter abundance in each sample was calculated based on the S/N values or intensities of the reporter ion quantifier node.

#### 2.6.4. Protein Expression Analysis

UniProtKB IDs were mapped to gene names using the UniProt ID mapping tool [11]. Proteins with high dropout rates (>0.5) and high standard deviations in scaled abundance (>100 or >5 according to their distribution) were excluded from further analysis. To identify differentially expressed proteins (DEPs) between the responder and non-responder groups, we used the limma R package paired test to compare pre-and post-treatment conditions, and DEPs were selected at *p* < 0.25 [12]. Differences in gene expression were tested under paired conditions. The log-scaled fold-change (log2(post)–log2(pre)) expression for each sample was calculated, and limma was performed between responders and non-responders. Next, gene set enrichment analysis (GSEA) was applied to identify pathways using the KEGG gene set of DAVID (*p* < 0.05) [13,14] (Appendix A). Because the GSEA results uncovered ‘phagosome’, and ‘ECM-interaction receptor’ pathways involved in cellular interaction, we investigated whether the protein expression of ligand receptors exhibited the difference. The ligand receptor pairs were obtained from a knowledge-based database and a single-cell lung cancer study [15]. We compared our DEPs with the ligand receptor collections using Venn analysis.

### 2.7. Flow Cytometry Analysis

The total numbers of various immune cells, including myeloid-derived suppressor cells (MDSC), cytotoxic CD8+T-cells, and regulatory T cells, were counted in the peripheral blood of the patients using flow cytometry. Fresh PBMCs were separated using HetasepTM (Stem Cell, Vancouver, BC, Canada) as described in the manufacturer’s protocol and stained for fluorescence-activated cell sorting (FACS) analysis. The antibody for each immune cell is listed in the following: APC-Cy7 labeled anti-human CD11b (BD Biosciences Clone: ICRF44), BV510 labeled anti-human CD33 (BD Biosciences Clone: WM53), BV786 labeled anti-human HLA-DR (BD Biosciences Clone: G46-6), BUV395 labeled Anti-human CD14 (BD Biosciences Clone:MP9), FITC labeled anti-human Lineage (BD Sciences), BUV labeled anti-human CD3 (BD Science Clone: UCHT1), PerCP-Cy5.5 labeled anti-human CD8 (BD Science Clone: RPA-T8), BV650 labeled anti-human CD4 (BD Science Clone SK3), BY711 labeled anti-human CD25 (BD Science Clone: 2A3), AF7647 labeled anti-FVS (BD Science Clone: Live/Dead), BUY496 labeled anti-human CD16 (BD Science Clone:3G8), AF647 labeled anti-human CD127 (BD Science Clone: RDR5), and PE-CF594 labeled anti-human CD15 (BD Biosciences Clone: W6D3). Stained cells were analyzed using Fortessa (BD Bioscience, Franklin Lakes, NJ, USA), and the data were analyzed using FlowJo (v10) software (FlowJo LLC, Vancouver, BC, Canada). MDSC subpopulation phenotypes were defined as described by Passaro et al. as follows: granulocytic MDSC, Lin-CD11b+CD33+CD11b+HLA-DR−/lowCD14−CD15+; monocytic MDSC, Lin-CD11b+CD33+CD11b+HLA-DR−/lowCD14+CD15−.

### 2.8. Validation Cohort Selection and Enzyme-Linked Immunosorbent Assay (ELISA) for TFRC

The validation cohort was composed of patients who had been treated with anti-PD1 or-PD-L1 antibodies between April 2016 and December 2022. Clinical data, including patient age, gender, smoking status, ECOG PS, line of therapy, PD-L1 expression level, PD-1 inhibitor type, and response to immunotherapy, were retrospectively analyzed. To assess the TFRC level of each patient, plasma was obtained by placing whole blood into a tube containing 0.5 m EDTA. The samples were centrifuged at 3000 rpm, 4 °C for 15 min. Plasma was collected and stored at −80 °C until further processing. TFRC concentrations were detected using ELISA kits (Thermo Fisher Scientific, Waltham, MA, USA) according to the manufacturer’s protocols.

### 2.9. Statistical Methods

A sample size (n = 18) was calculated to provide 80% power to demonstrate that the best ORR exceeded 10% at a one-sided type I error rate of 10% when the expected ORR in the treatment group was 30%. A total of 22 patients were planned for enrollment, considering a drop-out rate of 15%. The safety analysis population included all patients who received at least one dose of the study drug, and efficacy analyses were performed on the intention-to-treat population. Standard statistical methods were used, including descriptive statistics, the ꭓ^2^ test, logistic regression, the Kaplan–Meier method, two-sided 95% confidence interval (CI), and the stratified log-rank test. Data analyses were performed using SPSS (version 25.0; IBM Corporation, Armonk, NY, USA) and GraphPad Prism (version 9.5; GraphPad Software, San Diego, CA, USA).

## 3. Results

### 3.1. Patients Characteristics

A total of 22 patients were assigned to the treatment group and underwent efficacy and safety analyses. The patient demographics and clinical characteristics are listed in Table 1. The median age of the patients was 63 years. Most of the patients were male (82%), former or current smokers (82%), had an Eastern Cooperative Oncology Group performance status of 1 (86%), and had adenocarcinoma histology (73%). PD-L1 expression was negative (<1%) in 64% of the patients. Seventeen of the patients were enrolled in this study within three months of their last treatment. Six of the patients showed PD as the best response to prior treatment.

### 3.2. Treatment Efficacy

The median PFS and OS for the total patients were 2.3 months (95% CI: 1.3–3.2) and 11.6 months (95% CI: 6.2–17.0), respectively. (Table 2, Figure 1A,B).

When comparing survival according to the PD-L1 expression level and TMB, the median PFS for patients with negative PD-L1 (<1%) (n = 14) was 2.3 months (95% CI: 1.0–3.5), and that for those with a PD-L1 expression of ≥1% and <10% (n = 8) was 2.1 months (95% CI: 1.0–3.5, Figure 1C). The median PFS for patients with high TMB (n = 7) was 1.4 months (95% CI: 1.3–1.4), and that for those with low TMB (n = 10) was 2.3 months (95% CI: 1.7–2.9, Figure 1D). The median OS for patients with negative PD-L1 (<1%) (n = 14) was 13.5 months (95% CI: 10.5–16.4), and that for those with a PD-L1 expression of ≥1% and <10% (n = 8) was 8.7 months (95% CI: 5.1–12.4, Figure 1E). The median OS for patients with high TMB (n = 7) was 16.8 months (95% CI: 0.0–37.5), and that for those who had low TMB (n = 10) was 9.0 months (95% CI: 3.6–14.4, Figure 1F). The median PFS and OS for patients with large-cell neuroendocrine carcinoma (n = 5) was 1.4 months (95% CI: 1.3–1.5) and 1.4 months (95% CI: 1.3–1.4), respectively, and none of them showed a response (one SD and four PD).

The tumor burden change in reference to the baseline at the best overall response (BOR) ranged from -83.7% to +115.4% (median: +6.7%). Two patients achieved PR; thus, the ORR was 9% (95% CI, 1.9–26.1). The BOR in the remaining patients was SD in 11 patients (50%) and PD in nine patients (41%) (Figure 2A).

### 3.3. Tumor Burden Dynamics and Association with Survival

The tumor burden dynamics during treatment are shown in a spider plot (Figure 2B). Two patients with PR remained responsive even after discontinuation of treatment. The tumor burden decreased in five patients (22.7%). Five patients with decreased tumor burden had longer OS (24.8 months [95% CI, 5.9–43.7]) compared to those who only had an increment of tumor burden (9.0 months [95% CI, 4.3–13.7]) (*p* = 0.016) (Figure 2C).

### 3.4. Blood NGS Analysis

Mutation profiles obtained using blood NGS are summarized in Figure 2A. Although we enrolled patients with wild-type EGFR and ALK, one patient with the EGFR L858R mutation was identified using plasma NGS. KRAS and BRAF mutations were identified in 6 and 2 patients, respectively. Additionally, an EGFR exon 20 insertion was identified. Among the 22 patients, seven had TMB-high tumors, 10 had TMB-low tumors, and in the remaining five patients, TMC could not be assessed.

### 3.5. MDSC Analysis in Peripheral Blood

MDSCs have phenotypes similar to immature myeloid cells, which differentiate into dendritic cells, macrophages, and granulocytes. However, MDSCs can accumulate abnormally in the blood under chronic inflammatory conditions such as in cancer [16] and can inhibit antitumor immunity [17]. In a previous report, MDSC in patient blood was reported to be inversely correlated with therapeutic response [18]. To address the relevance of MDSCs, the percentages of total MDSC, granulocytic MDSC (gMDSC), and monocytic MDSC (mMDSC) in the peripheral blood were determined based on cell-surface markers. As shown in Figure 3A, all the MDSCs were separated using surface expression markers, two different MDSC populations (gMDSC and mMDSC) were plotted as the ratio of MDSCs to CD8+ cells, and two patients with PR maintained the lowest ratio throughout the treatment (Figure 3B,C). Furthermore, when we compared the changing ratios of MDSCs to CD8+ cells between the baseline and the last treatment, two patients with PR maintained the lowest ratio (Figure 3D), suggesting that the efficacy of cyclophosphamide and adriamycin induction therapy followed by nivolumab is correlated with decreased MDSC.

### 3.6. Protein Expression Analysis

In total, 1513 proteins were identified in 20 pre- and post-treatment paired samples. To identify the biological functions of the responders, we extracted differentially-expressed proteins (DEPs) to compare the responder and non-responder groups for three cases of pre- and post-treatment, as well as the fold change (FC) of paired samples (Figure 4A). Finally, 236–284 up and downregulated DEPs were identified using Limma (*p* < 0.25). Next, we investigated the pathways enriched in the DEPs using gene set enrichment analysis (GSEA) (*p* < 0.05). In the pre-treatment condition, upregulated DEPs of the responder group were involved in the ‘ECM-receptor interaction’ and ‘phagosome’ pathways. From proteins up-expressed in post-treatment, ‘phagosome’, ‘ECM-receptor interaction’, and ‘apoptosis’ pathways consistently emerged. To verify the paired upregulation status, we performed GSEA using the difference in protein expression fold changes (log2(post/pre-treatment)) between the responder and non-responder groups. Consistently, ‘phagosome’, ‘Fc gamma R-mediated phagocytosis’, ‘ECM-receptor interaction’, and ‘TGF-beta signaling’ pathways were detected. Therefore, responders were persistently altered in pathways associated with cell–cell interactions and phagocytosis.

To verify cell-interaction-associated pathways, we investigated whether known ligand receptor–protein interactions were detected in the responder DEPs. Previously published lung cancer single-cell landscapes portrayed ligand receptor interactions between tumor microenvironment cells and tumor cells from primary and metastatic tumors [15]. To investigate these gene interactions, we collected lung-cancer-specific ligand receptor pairs from a single-cell study and compared our DEPs with known lung cancer ligand receptor pairs using a Venn diagram (Figure 4B). We identified four proteins that exhibited dramatic change in expression levels in responders. Among them, TGFB1, THBS1, and CCL5 showed low abundance in the pre-treatment samples and were elevated after treatment. Meanwhile, TFRC levels were higher in responders compared to non-responders at baseline (fold change > 1.2) but decreased significantly in responders after treatment. (Figure 4C).

### 3.7. Validation of Predictive Value of TFRC in an Independent Cohort

To validate the predictive role of TFRC in patients with low PD-L1 expression (<10%), we analyzed an independent cohort, composed of 167 patients who were treated with anti-PD1 or-PD-L1 antibodies (Appendix A). First, we classified the patients according to PD-L1 expression level (n = 164). As expected, patients with low PD-L1 expression (<10%) showed significantly shorter OS and PFS compared to patients with high PD-L1 expression (≥10%) (15.2 months versus 12.4 months, *p* = 0.002 for OS; 2.5 months versus 1.4 months for PFS, *p* < 0.001 for PFS) (Figure 5A-1,A-2). Next, we conducted ELISA for TFRC in patients who were available for plasma samples (n = 50); the median OS for patients with high TFRC level (n = 25) and low TFRC level (n = 25) was 44.4 months and 7.3 months, respectively (hazard ratio [HR] 0.52, *p* = 0.065) (Figure 5B-1). The median PFS for patients with high TFRC level and with low TFRC level was 2.2 months and 1.6 months, respectively (HR 0.98, *p* = 0.95) (Figure 5B-2). We then focused on the patients with less than 10% of PD-L1 expression, which was same as the inclusion criteria of our trial. The median OS for these patients with high TFRC level (n = 13) and low TFRC level (n = 8) was 16.2 months and 5.8 months, respectively (HR 0.3, *p* = 0.036) (Figure 5C-1). The median PFS for the patients with high TFRC level was 1.3 months, and that for those with low TFRC level was 1.1 months (HR 0.5, *p* = 0.12) (Figure 5C-2). Lastly, we performed Cox regression analysis and found high TFRC level as an independent factor related to better OS and PFS (Figure 5D-1,D-2).

This section may be divided by subheadings. It provides a concise and precise description of the experimental results, their interpretation, and the experimental conclusions that can be drawn.

## 4. Discussion

Currently, IHC-based PD-L1 expression is the only approved predictive marker for anti–PD-1 therapy, and its accuracy is insufficient to discriminate responders from non-responders completely. Among the immunosuppressive mechanisms that weaken the ICI response, suppressive immune regulatory cells such as Tregs and MDSCs play a key role in promoting tumor progression and inhibiting adaptive and innate immunity [19,20]. Therefore, targeting these suppressive mechanisms is essential for maximizing the efficacy of ICIs. Our study showed that two patients with a durable response maintained the lowest ratio (<1) of MDSC to CD8+T-cell throughout the treatment, consistent with previous features. However, our results also indicated that most patients had no effect on CA induction patients, which is different from the expectations of previous in vivo studies. Our results may be explained by other baseline characteristics. In an early survival analysis of patients with CM 057, Peters et al. showed that patients with poor prognostic characteristics (<3 months since the last treatment, progressive disease as the best response to prior treatment, ECOG performance status of 1), together with low or no PD-L1 expression, were at a higher risk of death within the first 3 months of treatment with nivolumab than with docetaxel [2]. Therefore, the fact that most of the patients in our study had an ECOG status of 1 and less than 3 months after the last treatment may have led to a poor prognosis.

Innate immune checkpoints, which prevent malignant cells from being detected and removed through phagocytosis and suppress innate immune sensing, are also crucial for tumor-mediated immune escape and might be potential targets for cancer immunotherapy. Our study revealed that the expression levels of some proteins associated with phagocytosis were related to this response. Based on a previous study, TGFB1, known to interact with TGF-beta receptor 2, was involved in the interaction between metastatic lung cancer cells with mo-Macs (monocyte-macrophage). CCL5 interacts with exhausted CD8+ T cells and mo-Mac cells across primary lung tumors and metastases. THBS1 was involved in alveolar macrophage, interacting with the a3b1 complex of cancer cells. Meanwhile, TFRC expressed in malignant cells participates in the interaction between mo-Macs and malignant cells, including precancerous cells. Four proteins implied that three proteins of Mo-Mac cells were activated post-treatment, but the TFRC of the tumor cells decreased. Therefore, the results indicated that the treatment effectively inhibited tumor cells in responder patients.

TFRC can be used to predict immune phenotypes and immune cell infiltration in pancreatic cancer [21]. TFRC is positively associated with many immunomodulators and is co-expressed with several significant immune checkpoints. Ferroptosis is a recently discovered form of regulated cell death characterized by the accumulation of lipid peroxides and subsequent destruction of cellular membranes [22,23]. The role of the TFRC gene and its protein product, the transferrin receptor, is responsible for iron uptake by cells, and iron is a key regulator of ferroptosis. Iron is necessary for ROS generation of reactive oxygen species, which can promote lipid peroxidation, ultimately leading to ferroptosis. TFRC plays a role in regulating iron homeostasis, and the overexpression of this gene can lead to increased iron uptake and susceptibility to ferroptosis [24]. Studies have shown that ferroptotic cells can release damage-associated molecular patterns recognized by immune cells and trigger an immune response against cancer cells. These results indicate that the expression level of TFRC can affect the tumor-immune microenvironment, providing a new reference for the prognosis of ICI treatment [25]. To the best of our knowledge, this is the first study to prove TFRC level as a predictive marker for anti-PD-1 therapy response in NSCLC patients with low PD-L1 expression. However, caution is needed for this mechanism to be a potential target for cancer immunotherapy because it has been shown that in vitro treatment of macrophages with the transferrin receptor ligand, transferrin, can promote the M2 polarization state; in other words, it can also contribute to cancer progression and therapeutic resistance [26].

## 5. Conclusions

This trial had some limitations, including limited sample size and single-center design. At present, a combination of multiple agents is often used to overcome the lack of immune response in cold tumors and transform cold tumors into hot tumors, as did our study. Although CA IT did not much improve the efficacy of nivolumab in NSQ-NSCLC with PD-L1 expression < 10%, two patients showed dramatic and durable responses with decreasing MDSC, which contributed to poor outcomes with immunotherapy. Furthermore, we identified TFRC level as a predictive biomarker for immunotherapy in patients with low PD-L1 expression. Further large studies are needed to assess the predictive role of TFRC as a biomarker for ICIs in NSQ-NSCLC with low PD-L1 expression.

## Figures and Tables

**Figure 1 cancers-15-04460-f001:**
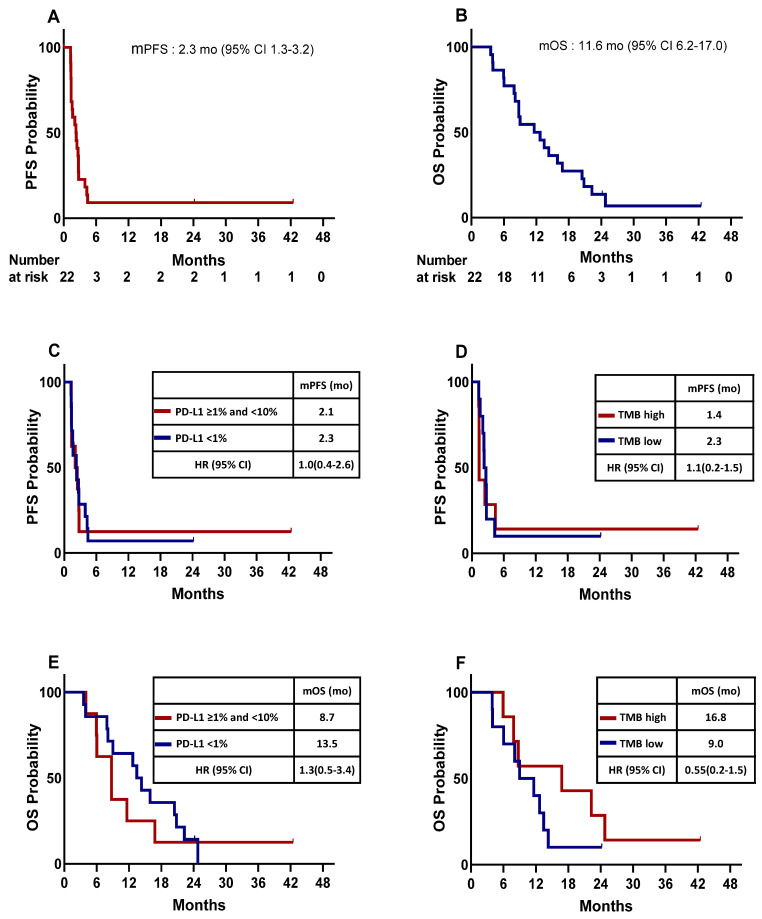
Kaplan–Meier plot for the median progression-free survival (mPFS) and median overall survival (mOS) calculated from the date of the beginning of enrollment. (**A**) mPFS for the entire study population. (**B**) mOS for the entire study population. mPFS of patients stratified by PD-L1 expression (**C**) and tumor mutation burden (TMB) (**D**). mOS of patients stratified by PD-L1 expression (**E**), TMB (**F**). HR, hazard ration; CI, confidence interval; mo, month; PFS; OS.

**Figure 2 cancers-15-04460-f002:**
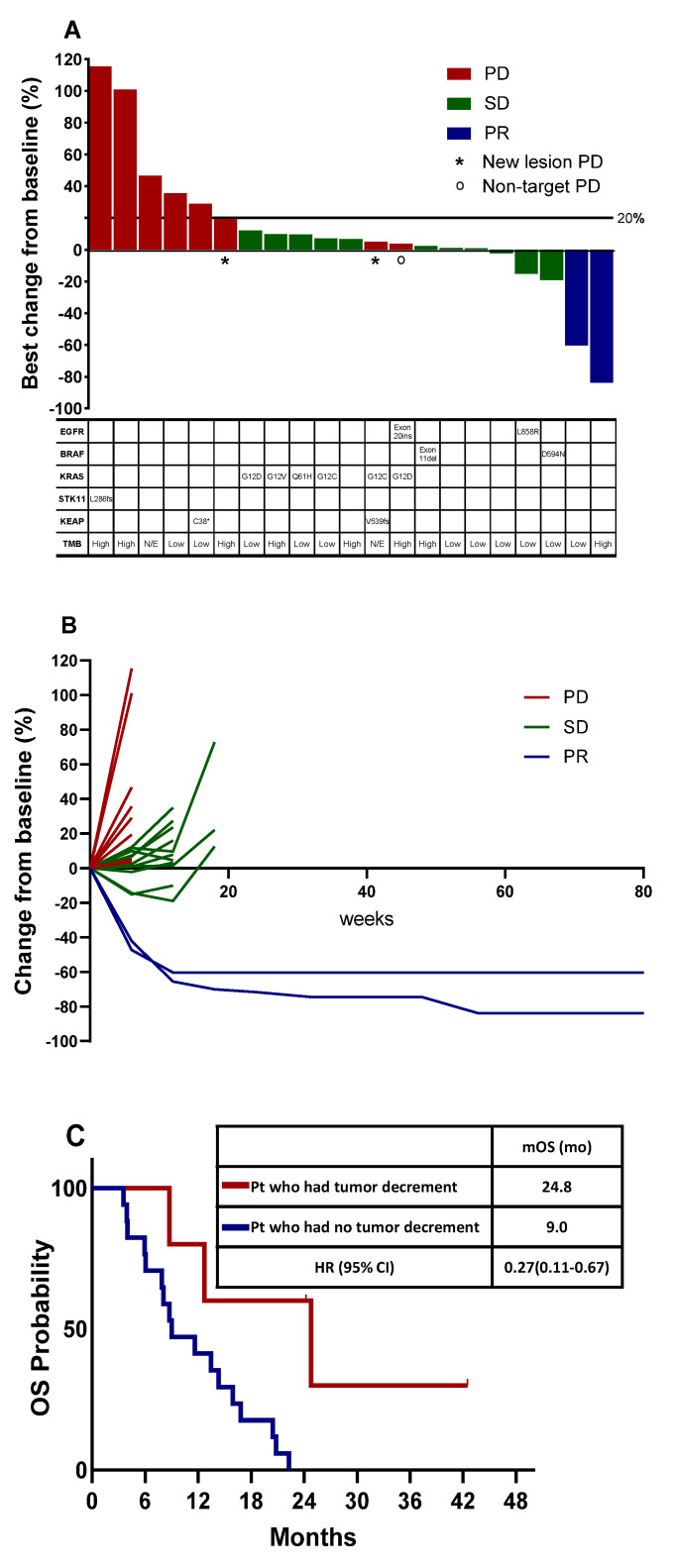
Tumor burden dynamics and association with survival. (**A**) Waterfall plot of best percentage change from baseline in target lesion size by best overall confirmed response, and mutation profiles obtained using blood next-generation sequencing. (**B**) Spider plot of tumor burden changes during the treatment. (**C**) Overall survival in the total population stratified by tumor burden dynamics. PD, progressive disease; SD, stable disease; PR, partial response; mOS, median overall survival; HR, hazard ration; CI, confidence interval; mo, month.

**Figure 3 cancers-15-04460-f003:**
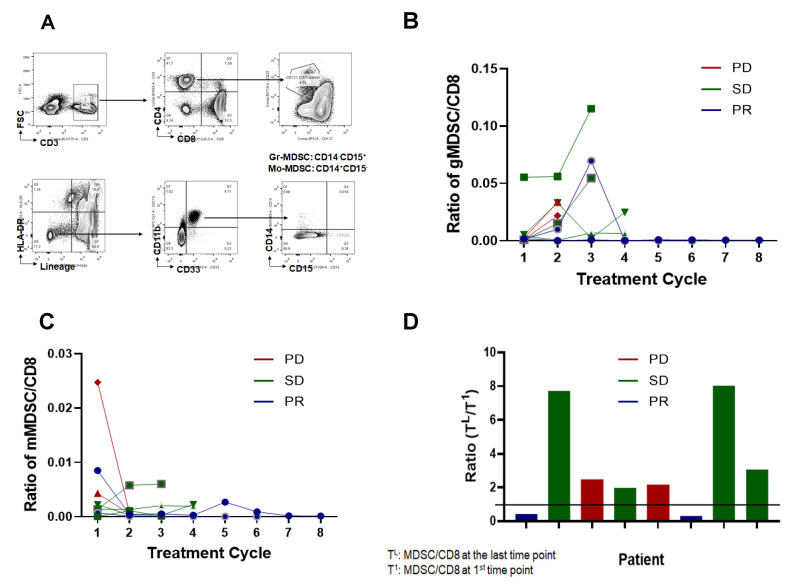
Detection of myeloid-derived suppressor cells (MDSC) in patient plasma. (**A**) A representative gating strategy. Fresh whole blood (WB) was placed for flow cytometry. CD45+ cells were selected; subsequently, T and B cells were excluded by gating on cells negative for anti-CD3, anti-CD19, and anti-CD20 antibodies. NK cells were excluded by gating on CD56− cells, and HLA-DR− cells were selected. Granulocytic MDSC was defined as Lin-CD11b+CD33+CD11b+HLA-DR−/lowCD14−CD15+ and monocytic MDSC was as Lin-CD11b+CD33+CD11b+HLA-DR−/lowCD14+CD15−. (**B**) Total number of gMDSC in 1 mL plasma counted and normalized with CD8+ T-cells. (**C**) Total number of mMDSC in plasma counted and normalized with CD8+ T-cells. (**D**) Changes of MDSC:CD8+ ratio during treatment, the ratio of MDSC/CD8+ at the indicated time points was divided with the initial time point. PD, progressive disease; SD, stable disease; PR, partial response.

**Figure 4 cancers-15-04460-f004:**
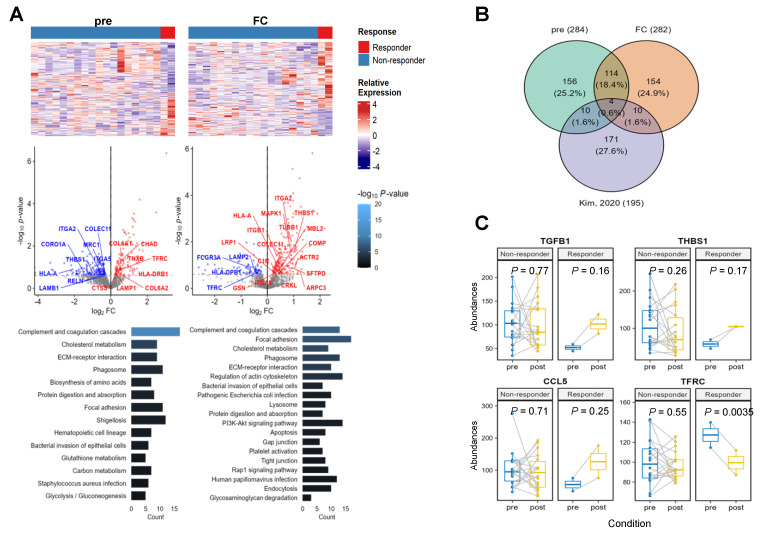
Protein expression analysis. (**A**) Differentially expressed proteins (DEPs) to compare responder and non-responder groups for pre-treatment as well as the fold change (FC) of paired samples. (**B**) A Venn diagram to extract known interaction ligand receptors between tumor microenvironment and lung tumor cells from our dataset. (**C**) Abundance change of four proteins in the intersection of Venn diagram.

**Figure 5 cancers-15-04460-f005:**
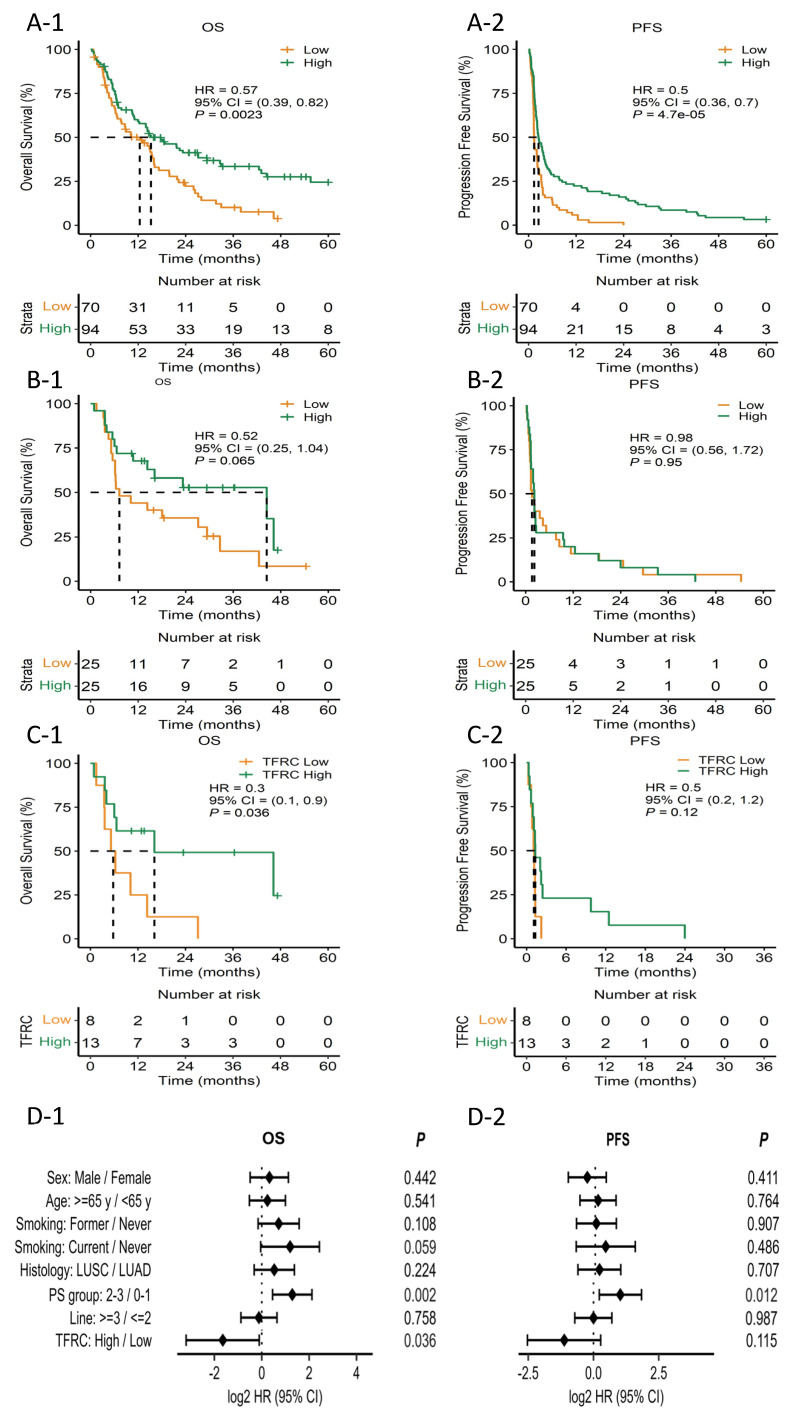
Kaplan–Meier plot for the median progression-free survival (mPFS) and median overall survival (mOS) calculated from the date of the beginning of immunotherapy. (**A-1**) mOS stratified by PD-L1 expression. (**A-2**) mPFS stratified by PD-L1 expression. (**B-1**) mOS stratified by TFRC expression. (**B-2**) mPFS stratified by TFRC expression. (**C-1**) mOS of patients stratified by TFRC expression in low PD-L1 patients. (**C-2**) mPFS of patients stratified by TFRC expression in low PD-L1 patients. (**D-1**) Forest plot for OS hazard ratio of patient factors and TFRC. (**D-2**) Forest plot for PFS hazard ratio of patient factors and TFRC. HR, hazard ration; CI, confidence interval; PS, performance status; PFS; OS; PD-L1; m; TFRC.

**Table 1 cancers-15-04460-t001:** Patient Demographics and Clinical Characteristics.

Characteristic	Total (N = 22)
**Age of diagnosis (years)**	
Median (range)	63 (37–72)
**Sex, n (%)**	
Male	18 (82)
Female	4 (18)
**Tobacco use, n (%)**	
Never	4 (18)
Former	11 (50)
Current	7 (32)
**ECOG PS, n (%)**	
0	3 (14)
1	19 (86)
**Previous line of therapy**	
1	16 (73)
2	6 (27)
**Tumor Histology, n (%)**	
Adenocarcinoma	16 (73)
Large cell neuroendocrine carcinoma	5 (23)
Sarcomatoid carcinoma	1 (4)
**PD-L1**	
<1%	14 (64)
≥1% and <10%	8 (36)
**Duration since the last treatment to study enrollment**	
<3 months	17 (77)
≥3 months	5 (23)
**Best response to prior treatment**	
PR	2 (9)
SD	11 (50)
PD	8 (36)
NE	1 (4)

ECOG PS, Eastern Cooperative Oncology Group performance status; PR, partial response; SD, stable disease; PD, progressive disease; NE, not evaluable; PD-L1, programmed death-ligand 1.

**Table 2 cancers-15-04460-t002:** Efficacy.

Response	Total (N = 22) n (%) [95% CI]
ORR, n (%) [95% CI]	2 (9) [1.9–26.1]
Complete response (CR)	0 (0)
Partial response (PR)	2 (9)
Stable disease (SD)	11 (50)
Progressive disease (PD)	9 (41)
DCR, n (%) [95% CI]	13 (59) [38.5–77.5]

ORR, objective response rate; CR, complete response; PR, partial response; SD, stable disease; PD, progressive disease; DCR, disease control rate; CI, confidence interval.

## Data Availability

The data presented in this study are available in this article and Appendix A.

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
