# Peer review of "Nivolumab after Induction Chemotherapy in Previously Treated Non-Small-Cell Lung Cancer Patients with Low PD-L1 Expression"

_cancers, 2023, doi:10.3390/cancers15184460_

Round 1

Reviewer 1 Report

I have carefully reviewed the manuscript titled "Nivolumab after Cyclophosphamide and Doxorubicin Induction Chemotherapy in Previously Treated Patients with Non-Squamous Cell Non-Small Cell Lung Cancer with Less than 10% of PD-L1 Expression." Overall, the study presents valuable insights into the treatment of non-squamous cell non-small cell lung cancer patients with low PD-L1 expression. However, there are certain areas that need to be addressed before the manuscript can be considered for publication. My main concerns are related to the clarity of the figures and the overall language of the manuscript.

Specific Comments:

  1. Figure Clarity: The figures provided in the manuscript, particularly Figure 2 and Figure 4, appear to lack the necessary resolution for clear data visualization. These figures are crucial for drawing meaningful inferences from the data presented. I kindly request that the authors resubmit these figures with a higher resolution to ensure better visibility and understanding of the results. Improved figures will undoubtedly enhance the overall quality of the manuscript.

  2. Language Polishing: While the manuscript's content is scientifically sound, there are instances where the language could be further polished to ensure better readability and comprehension. I recommend that the authors thoroughly review and revise the manuscript to enhance its overall clarity. This will greatly assist readers in understanding the methodology, results, and implications of the study.

Recommendations for Revision:

  1. Resubmit Figures: I strongly urge the authors to resubmit Figure 2 and Figure 4 with improved resolution to facilitate clear data interpretation. This will significantly contribute to the manuscript's scientific impact.

  2. Language Enhancement: The authors should carefully revise the entire manuscript to improve its language quality, ensuring that it is written in a clear, concise, and easily understandable manner. This will make the manuscript more accessible to a wider readership.

I appreciate the authors' dedication to advancing the understanding of treatment strategies for non-small cell lung cancer patients with low PD-L1 expression. Addressing the above-mentioned concerns will undoubtedly strengthen the manuscript's overall quality and contribute to its successful publication.

Sincerely,

Author Response

RESPONSE TO COMMENTS FROM REVIEWER #1

<Recommendations for Revision>

COMMENT #1

Resubmit Figures: I strongly urge the authors to resubmit Figure 2 and Figure 4 with improved resolution to facilitate clear data interpretation. This will significantly contribute to the manuscript's scientific impact

RESPONSE #1

Thank you for the insightful comment and sorry for the lack of the necessary resolution for clear data visualization. As reviewer suggested, we changed Figure 2 and Figure 4 with higher resolution.

COMMENT #2

Language Enhancement: The authors should carefully revise the entire manuscript to improve its language quality, ensuring that it is written in a clear, concise, and easily understandable manner. This will make the manuscript more accessible to a wider readership.

RESPONSE #2

We apologize for the incomprehensible English writing. As reviewer suggested we edited throughout the whole manuscript for better understanding.

Reviewer 2 Report

In this single group phase 2 study, the authors investigated the efficacy and safety of doxorubicin - cyclophosphamide, four cycles, associated with nivolumab starting at cycle 2 in a population of NSQ-NSCLC resistant to one or two previous lines of chemotherapy. Most of the patients have poor prognostic factors (less than 3 months since previous line, weak or lack of PD-L1 tissue score, etc.). The outomes (OR, PFS and OS) were quite disappointing. However, the authors demonstrated interesting data on MDSC / CD8+ ratio (decreasing in responders) and ferritin receptor expression (also decreasing in responders).

This is a well-written manuscript. Although the population tested is small, the comprehensive genetic, proteomic and immune phenotype analysis is impressive. The rationale of the doxo - Cyclo association is well defended in the introduction, and this is important because these cytotoxic drugs are not in use in NSCLC. The authors' intent was clearly to increase DAMPs and CD8+ in order to circumvent putative ICI resistance.

Few comments: 

The study would have taken more power with a two-group design for instance comparing  the AC-ICI with nivolumab alone or docetaxel alone. 

The hypothesis of this study (in term of alpha, power, population size calculation) should be indicated (why 22 patients only?). 

The 2015 WHO lung tumor pathologic classification does not any longer consider neuroendocrine large cell carcinoma (five patients in this study) as belonging to  NSCLC. They are now classified in high grade neuroendocrine tumor of the lung. The authors should at least give the outcome of this subgroup (usually NELCC are considered as resistant to ICI). 

Author Response

RESPONSE TO COMMENTS FROM REVIEWER #2

In this single group phase 2 study, the authors investigated the efficacy and safety of doxorubicin - cyclophosphamide, four cycles, associated with nivolumab starting at cycle 2 in a population of NSQ-NSCLC resistant to one or two previous lines of chemotherapy. Most of the patients have poor prognostic factors (less than 3 months since previous line, weak or lack of PD-L1 tissue score, etc.). The outomes (OR, PFS and OS) were quite disappointing. However, the authors demonstrated interesting data on MDSC / CD8+ ratio (decreasing in responders) and ferritin receptor expression (also decreasing in responders).

This is a well-written manuscript. Although the population tested is small, the comprehensive genetic, proteomic and immune phenotype analysis is impressive. The rationale of the doxo - Cyclo association is well defended in the introduction, and this is important because these cytotoxic drugs are not in use in NSCLC. The authors' intent was clearly to increase DAMPs and CD8+ in order to circumvent putative ICI resistance.

<Recommendations for Revision>

COMMENT #1

The study would have taken more power with a two-group design for instance comparing the AC-ICI with nivolumab alone or docetaxel alone.

RESPONSE #1

Thank you for the insightful comment. At the time when this trial was designed, nivolumab was a standard of care for pretreated advanced non–small cell lung cancer (NSCLC) which showed better efficacy over docetaxel in CheckMate057 trial. Therefore, the purpose of the study was to show improvement of nivolumab efficacy in low PD-L1 expressed patients after induction chemo, not to compare with cytotoxic chemotherapy.

COMMENT #2

The hypothesis of this study (in term of alpha, power, population size calculation) should be indicated (why 22 patients only?).

RESPONSE #2

First, a sample size of 18 evaluable patients came out to provide 80% power to demonstrate that the best ORR exceeded 10% at one-sided type I error rate of 10% when the expected ORR in the treatment group is 30%. Total of 22 patients was calculated from considering the drop-out rate of 15%.

We added this information to page 5, 2.9 Statistical Methods section.

A sample size (n=18) was calculated to provide 80% power to demonstrate that the best ORR exceeded 10% at one-sided type I error rate of 10% when the expected ORR in the treatment group is 30%. Total of 22 patients was planned for enrollment considering the drop-out rate of 15%.

COMMENT #3

The 2015 WHO lung tumor pathologic classification does not any longer consider neuroendocrine large cell carcinoma (five patients in this study) as belonging to NSCLC. They are now classified in high grade neuroendocrine tumor of the lung. The authors should at least give the outcome of this subgroup (usually NELCC are considered as resistant to ICI).

RESPONSE #3

Thank you for the precious comment. We analyzed the outcome of this NELCC subgroup.

The median PFS and OS for patients with NELCC (n=5) was 1.4 months (95% CI:1.3–1.5), and was 1.4 months (95% CI:1.3–1.4) respectively and none of them showed response (one SD and four PD).

We added this information to page 7, 3.2. Treatment efficacy. section

The median PFS and OS for patients with Large cell neuroendocrine carcinoma (n=5) was 1.4 months (95% CI:1.3–1.5), and was 1.4 months (95% CI:1.3–1.4) respectively and none of them showed response (one SD and four PD).

Reviewer 3 Report

The manuscript “Nivolumab after cyclophosphamide and doxorubicin induction chemotherapy in previously treated patients with non-squa-mous cell non-small cell lung cancer with less than 10% of PDL1 expression“ shows a somewhat better response of some patients with lung cancer who express TFRC treated with Nivolumab.  Although the results seem favorable the prognosis is still poor, this experience dealing with molecular markers is useful in the search of therapeutic improvements.  I think that a brief comment of cost-benefit would also be useful (e.g. P5,p2).  Also the title is too long, it could be shortened without missing relevant information.

No comments.

Author Response

RESPONSE TO COMMENTS FROM REVIEWER #3

The manuscript “Nivolumab after cyclophosphamide and doxorubicin induction chemotherapy in previously treated patients with non-squa-mous cell non-small cell lung cancer with less than 10% of PDL1 expression“ shows a somewhat better response of some patients with lung cancer who express TFRC treated with Nivolumab.  Although the results seem favorable the prognosis is still poor, this experience dealing with molecular markers is useful in the search of therapeutic improvements.

<Recommendations for Revision>

COMMENT #1

I think that a brief comment of cost-benefit would also be useful (e.g. P5,p2)

RESPONSE #1

Thank you for the important comment.

We added the comment in the manuscript page2 introduction section

 These two drugs are conventional cytotoxic drugs with good accessibility and cost benefits.

COMMENT #2

Also the title is too long, it could be shortened without missing relevant information.

RESPONSE #2

Thank you for the relevant comment. We shortened the title to below.

‘Nivolumab after induction chemotherapy in previously treated non-small cell lung cancer patients with low PD-L1 expression’